# The Effect of College Students’ Physical Activity Level on Depression and Personal Relationships

**DOI:** 10.3390/healthcare9050526

**Published:** 2021-04-29

**Authors:** Chang-Hwan Kim, Young-Eun Song, Yoo-Jeong Jeon

**Affiliations:** 1Department of Health and Exercise Management, Daejeon University, Daejeon 300-716, Korea; epckim@dju.kr; 2Department of Sports Science, Sun Moon University, Asan 336-708, Korea; jyjong20@sunmoon.ac.kr

**Keywords:** physical activity, depression, personal relationship

## Abstract

Background: Physical activity greatly affects human physical and mental health. This study investigated the effect of college students’ physical activity levels on depression and personal relationships. Methods: Participants were 525 college students from five Korean cities. The International Physical Activity Questionnaire, Beck’s Depression Inventory, and Leary’s interpersonal orientation paper test measured physical activity volume, depression, and interpersonal relationships, respectively. Data were analyzed using descriptive statistics, reliability analysis, one-way analysis of variance (ANOVA), and Pearson’s correlation analysis. Results: The results revealed significant differences among emotional, cognitive, and synchronous symptoms of depression across activity level groups. Regarding interpersonal relationships according to physical activity, for the sociometric disposition, there were differences between groups in the sympathetic-acceptable and sociable-friendliness factors and, for the expressive disposition, in the competitive-aggressive and rebellious-distrustful factors. Conclusions: There were statistical correlations between the physical activity volume and depression and the physical activity and interpersonal relationships. Subsequent research should examine college students’ physical activities and causal relationships among various psychological variables.

## 1. Introduction

University students experience various social, environmental, and psychological changes as they move from high school to university during the transition from adolescence to adulthood. They are especially vulnerable to mental health problems because they face various stressors such as academic problems, personal relationships, career and employment, and opposite-sex relationships [1,2,3,4]. An important stressor they may face is an inability to cope properly with the sudden increase in personal relationships.

University is a time when people socialize and learn to develop relationships; however, personal relationships are not easy to maintain. Interpersonal skills are very important for university students who wish to become social workers after graduating. Although there are individual differences, difficulties arising from relationships with friends, seniors, juniors, and professors can cause disappointment, loss of self-esteem, and fear of other people’s judgment, which may lead to avoiding personal relationships [5]. In addition, interpersonal relationship difficulties can cause social incompatibility and depression accompanied by communication difficulties, which may result in suicide [5,6,7,8,9]. We should pay attention to the CDC (Centers for Disease Control)’s announcement that depression in adolescence is a major predictor of depression after middle age [10], and the CDC announced that adolescents (18–24 years) are more depressed than adults are [11]. Therefore, it is important to find various ways to improve the mental health of university students [12].

Regular physical activity lowers the risk of a variety of conditions, including coronary artery diseases, heart diseases, stroke, diabetes, high blood pressure, colorectal cancer, breast cancer, and overweight and obesity. It also improves mental health and the quality of life [13]. However, the WHO stated that worldwide, about 23% of the adult population and 81% of teenagers attending school are not active enough, with leisure time reflecting a decreased participation in physical activities, as people spend more time sitting down due to various factors, including social and environmental [14]. In addition, various studies have shown that physical activity decreases during the transition from adolescence to adulthood [15,16]. These problems can be improved through regular and appropriate physical activities. In a recent previous study, as a result of a study on the association between physical activity and depression symptoms in adolescents, it was found that those who were active physically were less likely to develop symptoms of depression and that depression played an important role in social interactions [17]. In addition, a cross-sectional study of the European youth found that the lowest levels of depression and anxiety were associated with the positive mental health of students participating in team sports [18]. Additionally, according to a longitudinal study of Canadian adolescents, a report that showed decreased symptoms of depression when a part of a sports team looked at the relationship between physical activity and mental health [19]. Exercise improves physical ability, generates positive physical perception, and promotes psychological well-being. Additionally, sports involvement among adolescents helps them understand other’s perspectives and influences their interpersonal relationships positively [20].

Due to the difficulty in maintaining personal relationships among university students, it is necessary to study the relationship between depression and interpersonal tendencies in university students. The formation of good relationships becomes an essential element of social functioning and emotional and psychological well-being. Regular physical activity also promotes and improves psychological well-being [21]. Paying attention to the attitudes, frequency, time, and intensity of university students’ participation in physical activities and after-school programs can improve their mental health [22]. Therefore, it is necessary to compare and analyze the difference between the amount of physical activity, depression, and interpersonal relations among college students and conduct a study on whether the amount of physical activity can predict depression and interpersonal relations. Through this study, I think it can be used as empirical information necessary to establish an intervention strategy to prevent depression and improve interpersonal relations while improving the low physical activity of Korean university students.

Therefore, the purpose of this study was to provide basic data for managing depression and interpersonal tendencies through physical activity by investigating the amount of physical activity of college students and analyzing whether there was a difference in depression and interpersonal relationship propensity according to the amount of physical activity.

The hypothesis, according to the research purpose, was set up as follows.

First, there was a difference in depression according to the amount of physical activity of college students.Second, there was a difference in interpersonal relationship propensity according to the amount of physical activity of college students.Third, there were correlations between the amount of physical activity, depression, and interpersonal relationships among college students.

## 2. Materials and Methods

### 2.1. Study Participants

The study was conducted as a one-time survey by convenience sampling with a cross-sectional research design. The study was conducted with students attending five universities in four cities. The survey was conducted by researchers and researchers with research experience visiting the university. First, the professors who were lecturing at the university were asked for cooperation in this study, and then the students were informed of the purpose and procedure of the research on the questionnaire survey. Therefore, only students who agreed to voluntarily participate were asked to fill out the self-assessment writing method, and when the questionnaire response was completed, it was immediately collected at the site. Out of the total 600 respondents, 525 surveys were used in the analysis. Exclusion criteria included surveys deemed to have insincere or incomplete responses. Based on these criteria, 75 participants were excluded. Demographic information collected included sex, grade, and continuous participation in physical activity (more than 30 min at one time, perspiration level, three times a week or more, longer than six months), and physical activity levels (Table 1). This study was approved by the blinded for review in 2018 and investigated between 2019 and 2020 (approval number: 1040647-201812-HR-006). All participants provided written informed consent.

### 2.2. Research Tools and Reliability

#### 2.2.1. Amount of Physical Activity

The study measured physical activity with the Korean version of the Short Message Self-encoding questionnaire of the International Physical Activity Questionnaire (IPAQ) (www.ipaq.ki.se, 5 October 2020). It consisted of four areas: leisure time, activities at home or outdoors, activities related to work and movement, and documenting physical activities for seven days. For each area, the frequency (days/week) and time (min/day) were asked depending on the specific type of activity, such as low, medium, and high intensity. The collected data were used to calculate the amount of physical activity (in metabolic equivalents; MET-min/week) according to the formula (Table 2).

College students’ physical activity levels were classified into three groups: low-, medium-, and high-intensity activities, according to the degree of physical activity. The criteria for classification are shown (Table 3).

#### 2.2.2. Depression Level

The tool used to measure the level of depression was developed by Beck et al. [23] and adapted by Lee and Song [24]. This measure of depression consisted of 21 questions for identifying emotional, cognitive, synchronous, and physiological symptoms of depression, and each question was a self-reported item rated on a four-point scale ranging from zero to three. The total score ranges from 0–63, with scores of 9 or less, 10–15, 16–23, and 24–63 indicating no depression, mild depression, moderate depression, and severe depression, respectively. The specific areas of depression and reliability identified through Cronbach’s α coefficients are shown (Table 4).

#### 2.2.3. Personal Relationships

The tool used to measure personal relationships was a personal relationship test reconstructed by Ahn [25], referring to the personal relationship behavior model of Leary [26]. This personal relationship test was divided into three areas: role disposition, social-relational disposition, and expressive disposition, which in turn consisted of 84 items. Each question was measured with a four-point Likert-type scale, with responses ranging from one (*not entirely*) to four (*always*). A higher score indicated a highly personal relationship in the area. The specific areas of the personal relationship test and the reliability identified through Cronbach’s α coefficients are shown (Table 5).

### 2.3. Data Analysis

All data collected were entered into the SPSS 16.0 statistical program and analyzed according to the research objectives. Descriptive statistics were calculated to identify the demographic and physical characteristics of the study participants, and reliability analysis was conducted to verify the reliability of the questionnaires. In addition, a one-way analysis of variance (ANOVA) was performed to examine the level of depression and personal relations according to physical activity level. Post-hoc analysis was performed using Tukey’s test, and correlation analysis was conducted to examine the extent of correlations among physical activity level, depression, and personal relationships.

## 3. Results

### 3.1. Differences in Level of Depression According to Physical Activity

The level of depression among college students with emotional symptoms (*F* = 6.807, *p* = *0*.001), cognitive symptoms (*F* = 10.635, *p* = 0.001), synchronous symptoms (*F* = 8.397, *p* = 0.001) varied depending on the amount of physical activity they reported (Table 6). The post-hoc analysis found that depression with emotional symptoms was significantly higher in low-intensity physical activity groups than in medium- and high-intensity physical activity groups. Further, low-intensity physical activity groups had significantly higher levels of cognitive depression than medium- and high-intensity physical activity groups. Finally, it was found that low-intensity physical activity groups had statistically higher levels of synchronous depression than medium- and high-intensity physical activity groups.

### 3.2. Differences in Personal Relationships According to Physical Activity Levels

Each disposition in the three areas of personal relationships had sub-areas: the role disposition was divided into dominant-ascendant and independent-responsible; the sociometric disposition comprised sympathetic-acceptable and sociable-friendliness sub-areas; and the expressive disposition consisted of competitive-aggressive, display-self-absorption, and rebellious-distrustful.

According to the amount of physical activity, the personal relationship among college students was statistically significant in social relations for the sympathetic-acceptable (*F* = 12.118, *p* = 0.001) and sociable-friendliness (*F* = 22.699, *p* = 0.001) sub-areas (Table 7). Both of these factors in social relations indicated that high-intensity physical activity groups fared better than low-and medium-intensity physical activity groups. Finally, there was a statistically significant difference for the competitive-aggressive (*F* = 5.209, *p* = 0.006) and rebellious-distrustful (*F* = 3.143, *p* = 0.044) factors of the expressive disposition: high-intensity physical activity groups were scored higher than low-intensity physical activity groups.

### 3.3. Correlation between Physical Activity, Level of Depression, and Personal Relationships

It showed the correlations among students’ physical activity levels, depression levels, and personal relationships (Table 8). The results of the correlation analysis found that physical activity and depression were negatively correlated, such that high levels of physical activity were associated with low levels of depression.

Physical activity and personal relationships were found to have a statistically significant, statistically positive correlation in the dimensions of dominant-ascendant, sympathetic-acceptable, sociable-friendliness, competitive-aggressive, and display-self-absorption. In other words, greater levels of physical activity were associated with more positive personal relationships.

## 4. Discussion

Physical activity among college students has a positive effect on physical and mental health in adulthood. Therefore, this study aimed to analyze the differences in depression and personal relationships according to the amount of physical activity among college students and identify the relationships between these variables to provide preliminary data for managing depression and personal relationships through physical activities.

The study classified students into three groups based on activity level—low-, medium-, and high-intensity activities—to determine whether college students who engage in low or high physical activity were more likely to be depressed. The depression level of the low-intensity activity group was significantly higher than that of the medium- and high-intensity activity groups. The ANOVA showed statistically significant differences among groups for emotional symptom (*F* = 6.807, *p* = 0.001), cognitive symptom (*F* = 10.635, *p* = 0.001), synchronous symptom (*F* = 8.397, *p* = 0.001) depression.

These results were similar to those reported in a previous study of university students, which found that the high intensity of physical activities significantly reduced psychological pain such as anxiety and depression, and the high intensity of physical activities was more affected by men than women, especially on mental health indicators [27]. These results were similar to those reported in a previous study of underweight female university students that found that these students had the highest rates of emotional and physiological depression and were in the low-intensity physical activity group [28]. Another study found that physical activity had a positive effect on physical health and that depression levels and anxiety decreased due to physical activity or exercise above medium-intensity [29,30]. Prior studies have also shown that female university students’ participation in physical activities increased their self-esteem and reduced their level of depression [31,32]. Related to these studies, research by Kim and Bae [33] and Lee et al. [34] identified that the main causes of depression were primarily personal and social conditions, such as economic conditions, family, or social support in the case of female students. Hence, strategies for improving individual and social environments, along with methods for increasing physical activity, were deemed necessary to manage female students’ depression. In addition, physical activity programs for both male and female university students should not solely contain high-intensity exercises. The study found that college students with high levels of depression were in the low-intensity physical activity group and that the high-intensity physical activity group had the lowest depression. In other words, in order to manage college students’ depression through physical activities, it was deemed more effective to apply customized high-intensity and medium-intensity programs that were suitable for individual physical strength. 

Further, we found that college students’ personal relationships varied depending on the amount of their physical activity for some sub-factors. For sociometric disposition, sympathetic-acceptable (*F* = 12.118, *p* = 0.001), and sociable-friendliness (*F* = 22.689, *p* = 0.001) factors indicated that the personal relationships were higher in high-intensity than in low- and medium-intensity physical activity groups. For expressive disposition, the competitive-aggression (*F* = 5.209, *p* = 0.006) factor indicated that the level of personal relationships was higher in the high-intensity than in the low-intensity physical activity group. The rebellious-distrustful (*F* = 3.143, *p* = 0.044) dimension showed that the level of personal relationships was higher in the low-intensity than in the high-intensity physical activity group. In this study, the amount of physical activity positively affected the personal relationship in the high-intensity compared to the low- and medium-intensity groups.

These results partially supported the findings of Song and Kim [30], who reported that female students with high-intensity physical activity levels had positive personal relationships. The positive association with personal relationships supported the findings of a study by Suh and Yang [35], who reported that a high-intensity physical activity group of university students showed positive relationship factors. This was also in line with the study by Lee et al. [20] that reported that the continuous physical activity group’s physical self-concept, self-confidence, and satisfaction level were high in university students. Nicole and Toben’s [36] research on vigorous physical activity and mental health in college students supported our findings; they found that poor mental health and stress were less common among college students who exercised, and socialization partially mediated vigorous physical activity, mental health, and stress relationships. The difficulties of personal relationships experienced during university life could lead to social maladjustment and cause problems in establishing self-identity and character development. Therefore, while forming personal relationships was very important for university students, it was not easy to form and maintain amicable personal relationships. In this respect, the results of the study, which found positive personal relationships for college students in groups with high physical activity levels, had significant implications. The study showed that regular physical activities might positively change one’s personal relationships in the course of university life when it was difficult to maintain or form amicable personal relationships. Therefore, continuous efforts to increase the level of physical activity from low- to medium-, and from medium- to high-intensity were necessary to promote positive changes in personal relationships.

The present study found that physical activity in college students was negatively correlated with emotional, cognitive, and synchronous depression and positively correlated to the personal relationship sub-factors of dominant-ascendant, sympathetic-acceptable, sociable-friendliness, competitive-aggressive, and display-self-absorption. The more physical activity a participant engaged in, the less depressed they were, and the more positive their personal relationships were. In this regard, a study on the level of self-care, such as physical, mental, and personal relationships, found that physical activity participants were significantly higher in all sub-factors of self-care than non-participants [37], and many studies also showed that physical activity participation had a very strong correlation with mental health. Physical activity during university affected health and quality of life after graduation, and it was also closely related to mental health. It was necessary to identify the various stresses experienced by university students and seek positive countermeasures to mediate them. Adopting methods and systems, such as promoting university sports and requiring essential courses involving sports participation in liberal arts, were expected to positively impact the social adaptation and the preparation process for post-graduation life. The findings of this study suggested that requiring physical activity as a part of the college education might have beneficial effects on the physical and psychological health after graduation.

Summarizing the discussions in this study, it was imperative that college students participated in exercising as well as made efforts to increase the amount of active physical activity in their daily life. However, there was a limitation in that this study was conducted only for students enrolled in universities located in 4 cities out of a total of 17 cities and provinces in Korea, so there were some limitations to generalizing these results. In future research, I think it is necessary to derive research results for university students in more regions, taking into account different cultures and living environments in different regions. 

## 5. Conclusions

College students in Korea experience various social, environmental, and psychological changes in their transition to adulthood. This can cause depression and affect interpersonal relationships. It is important to collect basic data for managing university students’ depression and interpersonal relationships through physical activities, which are known to positively affect not only physical but also psychological health.

According to this study, first, college students’ depression was higher in groups with lower physical activity levels. Second, college students’ interpersonal relationships were more positive among people with higher physical activity levels. Finally, there was a statistical correlation between the amount of physical activity of a college student and their personality, as college students’ depression and interpersonal relationships were related to their physical activity. Therefore, controlling the amount of physical activity of university students might lower university students’ depression levels and positively impact their interpersonal relationships.

In conclusion, follow-up research needs to set the various psychological variables of university students as dependent variables and further verify the relationship between the physical activity volume and psychological factors. The period of physical activity was also expected to affect psychological changes, suggesting the need for experimental research to identify psychological changes depending on the amount and frequency of physical activity.

## Figures and Tables

**Table 1 healthcare-09-00526-t001:** Physical characteristics of the study participants (*n* = 525) CDC (Centers for Disease Control).

Characteristic	Sortation	Frequency	Percent
Sex	Male	259	49.3
	Female	266	50.7
Grade	1st Grade	172	32.8
	2nd Grade	164	31.2
	3rd Grade	107	20.4
	4th Grade	82	15.6
Continuous physical activity	Participation	215	41.0
	Nonparticipation	310	59.0
Physical activity level	Low intensity	171	32.6
	Medium intensity	205	39.0
	High intensity	149	28.4

**Table 2 healthcare-09-00526-t002:** Formula for calculating physical activity.

Category	Equation
Low intensity	3.3 × walking time (minutes) × walking days
Medium intensity	4.0 × medium intensity activity time (minutes) × days of high intensity activity
High intensity	8.0 × high intensity activity time (minutes) × days of high intensity activity
Total amount of physical activity	walking + medium intensity + high intensity

**Table 3 healthcare-09-00526-t003:** Physical activity level classification criteria.

**Sortation**	**Classification Standard**
Low intensity activity	Physical activity not applicable to high or medium intensity activities
Medium intensity activity	One of the three: In case high intensity physical activity (more than 8 METs per minute) was performed for more than 20 min a day at least 3 days a week. In case medium intensity physical activity (more than 4 METs per minute) was performed for at least 5 days a week for more than 30 min a day. In case of exercising more than 5 days for any combination of exercise and 600 MET-minutes per week or more
High intensity activity	One of the two: In case high intensity physical activity (more than 8 METs per minute) was performed at least 3 days a week for 1500 MET-minutes. In case combination of any level of exercise was more than 3000 MET-minutes a week by daily exercise.

MET = Metabolic Equivalents of Task (MET min/week).

**Table 4 healthcare-09-00526-t004:** Classification criteria for depressive symptoms.

Area	Number of Questions	Cronbach’s α
Emotional	6	0.781
Cognitive	5	0.720
Synchronous	5	0.724
Physiological	5	0.671
Total	21	0.897

**Table 5 healthcare-09-00526-t005:** Reliability of the personal relationship test.

Area	Factor	Number of Questions	Cronbach’s α
Role disposition	dominant-ascendant	9	0.706
	independent-responsible	11	0.689
Sociometric disposition	sympathetic-acceptable	11	0.718
	sociable-friendliness	14	0.799
Expressive disposition	competitive-aggressive	5	0.604
	display-self-absorption	8	0.624
	rebellious-distrustful	26	0.880
Total		84	0.915

**Table 6 healthcare-09-00526-t006:** Differences in depression according to the amount of physical activity (*n* = 525).

Factor	Physical Activity Level (*n*)	M ± SD	*F*	*p*	Post-Hoc
Depression with Emotional Symptoms	Low Intensity (171) ^a^	2.53 ± 2.47	6.807	0.001	a > b a > c
	Medium Intensity (205) ^b^	1.87 ± 2.16			
	High-Intensity (149) ^c^	1.64 ± 2.17			
Depression with Cognitive Symptoms	Low Intensity (171)	2.74 ± 2.61	10.635	0.001	a > b a > c
	Medium Intensity (205)	2.08 ± 2.17			
	High Intensity (149)	1.57 ± 2.00			
Depression with Synchronous Symptoms	Low Intensity (171)	3.10 ± 2.49	8.397	0.001	a > b a > c
	Medium Intensity (205)	2.45 ± 2.14			
	High Intensity (149)	2.10 ± 2.02			
Depression with Synchronous Symptoms	Low Intensity (171)	1.91 ± 2.57	1.623	0.198	
	Medium Intensity (205)	1.61 ± 1.72			
	High Intensity (149)	1.53 ± 1.78			
Total	Low Intensity (171)	10.29 ± 8.49	9.118	0.001	a > b a > c
	Medium Intensity (205)	8.03 ± 6.77			
	High Intensity (149)	6.85 ± 6.84			

Post-hoc: a = low intensity physical activity group, b = medium-intensity physical activity group, c = high intensity physical activity group.

**Table 7 healthcare-09-00526-t007:** Differences in personal relationships depending on the amount of physical activity (*n* = 525).

Factor		Group (*n*)	M ± SD	*F*	*p*	Post-Hoc
Role disposition	dominant-ascendant	Low Intensity (171) a	2.59 ± 0.38	2.805	0.061	
		Medium Intensity (205) b	2.59 ± 0.35			
		High Intensity (149) c	2.67 ± 0.36			
	independent-responsible	Low Intensity (171)	2.60 ± 0.36	1.063	0.346	
		Medium Intensity (205)	2.60 ± 0.36			
		High Intensity (149)	2.65 ± 0.35			
Sociometric disposition	sympathetic-acceptable	Low Intensity (171)	2.74 ± 0.33	12.118	0.001	a < c b < c
		Medium Intensity (205)	2.79 ± 0.37			
		High Intensity (149)	2.93 ± 0.35			
	sociable-friendliness	Low Intensity (171)	2.83 ± 0.31	22.699	0.001	a < c b < c
		Medium Intensity (205)	2.89 ± 0.35			
		High Intensity (149)	3.09 ± 0.36			
Expressive disposition	competitive-aggressive	Low Intensity (171)	2.68 ± 0.41	5.209	0.006	a < c
		Medium Intensity (205)	2.73 ± 0.40			
		High-Intensity (149)	2.83 ± 0.47			
	display-self-absorption	Low Intensity (171)	2.63 ± 0.38	2.040	0.131	
		Medium Intensity (205)	2.58 ± 0.36			
		High-Intensity (149)	2.66 ± 0.41			
	rebellious-distrustful	Low Intensity (171)	2.39 ± 0.38	3.143	0.044	a > c
		Medium Intensity (205)	2.33 ± 0.38			
		High-Intensity (149)	2.28 ± 0.39			

Post-hoc: a = low intensity physical activity group, b = medium-intensity physical activity group, c = high intensity physical activity group.

**Table 8 healthcare-09-00526-t008:** Correlation among physical activity, depression, and personal relationships.

Factor	1	2	3	4	5	6	7	8	9	10	11	12
1. Amount of physical activity	1											
2. Emotional	−0.141 ***	1										
3. Cognitive	−0.182 ***	0.737 ***	1									
4. Synchronous	−0.160 ***	0.733 ***	0.715 ***	1								
5. Physiological	−0.053	0.455 ***	0.453 ***	0.517 ***	1							
6. Dominant-ascendant	0.142 ***	−0.246 ***	−0.269 ***	−0.269 ***	0.011	1						
7. Independent-responsible	0.060	−0.104 *	−0.127 ***	−0.165 ***	0.042	0.675 ***	1					
8. Sympathetic-acceptable	0.223 ***	−0.150 ***	−0.140 ***	−0.165 ***	−0.018	0.475 ***	0.420 ***	1				
9. Sociable-friendliness	0.246 ***	−0.123 **	−0.156 ***	−0.154 ***	−0.043	0.395 ***	0.317 ***	0.658 ***	1			
10. Competitive-aggressive	0.104 *	−0.043	−0.059	−0.069	0.077	0.400 ***	0.371 ***	0.451 ***	0.413 ***	1		
11. Display-self-absorption	0.105 *	−0.064	−0.082	−0.159 ***	−0.002	0.510 ***	0.524 ***	0.401 ***	0.430 ***	0.373 ***	1	
12. Rebellious-distrustful	−0.075	0.043	0.026	−0.005	0.115 **	0.454 ***	0.531 ***	0.056	−0.133 ***	0.262 ***	0.467 ***	1

* *p* < 0.05, ** *p*< 0.01, *** *p* < 0.001.

## Data Availability

The data are not publicly available due to privacy or ethical reasons.

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
