# Peer review of "The Effect of College Students’ Physical Activity Level on Depression and Personal Relationships"

_healthcare, 2021, doi:10.3390/healthcare9050526_

Round 1

Reviewer 1 Report

This study brings a great novelty that is the relationship between personal relationships and the amount of physical exercise. Although the relationship between physical exercise and depression has been widely studied, the same does not happen with personal relationships. It was a pleasure to read this study. We very much appreciate the question of personal relationships.

We leave some suggestions for the authors to analyze if they are interested in adding/changing:

Summary

- check if they want to keep this information "and reliability analysis was conducted to verify the reliability of the questionnaires."; because these data were not presented in this paper. (see this information also in 2.3 data Analysis)

- it is not clear to us what the authors mean by “There were static correlations between”. Because of the “Static” concept. We did research, in an attempt to understand, but we continue with this doubt. Perhaps making this clearer can help other readers better understand your results.

Introduction

The authors present some percentages related to students and the level of physical activity (However, the WHO stated that worldwide, about 23% of the adult population and 81% of teenagers attending school are not active enough .. ") Suggestion: We understand that it was relevant to also present some percentages on the prevalence of depression in college students. The numbers give a real sense of the problem.

Material and methods

We believe that it would be relevant for the authors to include the date of the data collection. When do you start and end? (we noticed by the approval number: 1040647-201812-HR-006) that the avaliation possibly took place in 2018/2019, but this information could be clear.

It was a cross-sectional descriptive study, right? With a single moment of evaluation, right? Was sampling for convenience?

Ee think you can answer this question with a short sentence, which helps you to understand better.

Results:

The results mention: “The level of depression among university students, such as emotional (F = 6,807, p = .001), cognitive (F = 10,635, p = .001), and synchronous depression (F = 8,397, p =. 001 “ or “ Post-hoc analysis found that emotional depression was significantly… ”

However, we warn that there is no emotional depression, nor cognitive depression ...There are, yes, symptoms of depression from different domains (emotional, cognitive ...)

We understand the small confusion but we consider that it is very important to make this correction. Depression has emotional, cognitive symptoms ...; as the authors initially put it (2.2.2. Depression Level)

However, in the results, this information is not so clear. So it would look something like, e.g.: "Post-hoc analysis found that the emotional symptoms of depression… ” (see the other situations, in the discussion it is also described in the same way).

Point 3.2 Differences in Personal Relationships According to Physical Activity Levels. The paragraph: "Personal relationships are divided into three areas: role, ... s sub-areas; and expressive disposition."

It is unnecessary, it is already on top (2.2.3. Personal Relationships). The suggestion is to remove.

In point 3.3. Correlation between Physical Activity, Level of Depression, and Personal Relationships, the concept of static positive correlation appears again. Suggestion: clarify why it is Static.

Discussion

The authors state that “The present study found that university students with severe depression were the most physically active, and the lowest levels of depression were found among those in the medium-intensity physical activity group.” When do authors refer to “the present study"  do they refer to this study? The results do not support this data, right? It was what the authors observed ... ? it seems to us that this information is unclear. The results are " Low Intensity (171) 10.29±8.49. vs High Intensity (149) 6.85±6.84)

Some  details 

Introduction - Use the Acronym CDC without previously describing it

Method - it is recommended that Arabic numbers up to 10 be written in full (eg. “The study was conducted with students attending 5 universities in 4 cities.) The correct way is“ five universities in four cities). Please review the rest of the doc because there are other situations, like this one.

In point 6. Patents. Possibly want to mention - Author Contributions:

Author Response

" Please see the attachment "

Reviewer 2 Report

The manuscript is well-written and structured, and shows interesting findings for the readers of the journal. However, some improvements are recommended to perform a revision:

a) Further study rationale is necessary, as well as justification of the aim. Hypotheses should be described.

b) More complex data analysis is recommended, by applying multivariate analysis (a regression analysis, for example). Mediational or moderational analyses could be explored. 

c) Differences in the associations by gender and age may be examined.

d) Study limitations should be further developed.

e) Implications for practice and program design should be further developed.

Author Response

" Please see the attachment "

Round 2

Reviewer 2 Report

The suggested changes have all been performed. Thanks a lot for your effort.